# Does Avian Coronavirus Co-Circulate with Avian Paramyxovirus and Avian Influenza Virus in Wild Ducks in Siberia?

**DOI:** 10.3390/v15051121

**Published:** 2023-05-07

**Authors:** Kirill Sharshov, Nikita Dubovitskiy, Anastasiya Derko, Arina Loginova, Ilya Kolotygin, Dmitry Zhirov, Ivan Sobolev, Olga Kurskaya, Alexander Alekseev, Alexey Druzyaka, Pavel Ktitorov, Olga Kulikova, Guimei He, Zhenghuan Wang, Yuhai Bi, Alexander Shestopalov

**Affiliations:** 1Federal Research Center of Fundamental and Translational Medicine, Novosibirsk 630117, Russialoginova995@gmail.com (A.L.); dmitro.zhirov@yandex.ru (D.Z.); sobolev_i@hotmail.com (I.S.); kurskaya_og@mail.ru (O.K.); al-alexok@ngs.ru (A.A.); shestopalov2@mail.ru (A.S.); 2Faculty of Natural Sciences, Novosibirsk State University, Novosibirsk 630073, Russia; 3Institute of Animal Systematics and Ecology, Novosibirsk 630091, Russia; decartez@gmail.com; 4Institute of Biological Problems of the North, Magadan 685000, Russia; pktitorov@gmail.com (P.K.); gaerlach@gmail.com (O.K.); 5School of Life Sciences, East China Normal University, Shanghai 200062, China; gmhe@bio.ecnu.edu.cn (G.H.); zhwang@bio.ecnu.edu.cn (Z.W.); 6Center for Influenza Research and Early-warning (CASCIRE), CAS Key Laboratory of Pathogenic Microbiology and Immunology, Institute of Microbiology, Chinese Academy of Sciences, Beijing 100101, China; beeyh@im.ac.cn

**Keywords:** avian coronaviruses, *Gammacoronavirus*, *Deltacoronavirus*, ACoV, wild birds, co-circulation, avian influenza viruses, avian paramyxovirus, APMV, avian influenza surveillance, Siberia

## Abstract

Avian coronaviruses (ACoV) have been shown to be highly prevalent in wild bird populations. More work on avian coronavirus detection and diversity estimation is needed for the breeding territories of migrating birds, where the high diversity and high prevalence of *Orthomyxoviridae* and *Paramyxoviridae* have already been shown in wild birds. In order to detect ACoV RNA, we conducted PCR diagnostics of cloacal swab samples from birds, which we monitored during avian influenza A virus surveillance activities. Samples from two distant Asian regions of Russia (Sakhalin region and Novosibirsk region) were tested. Amplified fragments of the RNA-dependent RNA-polymerase (RdRp) of positive samples were partially sequenced to determine the species of *Coronaviridae* represented. The study revealed a high presence of ACoV among wild birds in Russia. Moreover, there was a high presence of birds co-infected with avian coronavirus, avian influenza virus, and avian paramyxovirus. We found one case of triple co-infection in a Northern Pintail (*Anas acuta*). Phylogenetic analysis revealed the circulation of a *Gammacoronavirus* species. A *Deltacoronavirus* species was not detected, which supports the data regarding the low prevalence of deltacoronaviruses among surveyed bird species.

## 1. Introduction

Avian coronaviruses are a non-taxonomic group of single-stranded (+) RNA viruses that infect wild and domestic birds. Avian coronaviruses are species of the *Coronaviridae* family, the *Orthocoronavirinae* subfamily of two genera (*Gammacoronavirus* and *Deltacoronavirus*). Based on International Committee on Taxonomy of Viruses (ICTV) reports, the *Gammacoronavirus* genus includes five species: *Goose coronavirus CB17* (*Brangacovirus* subgenus), *Beluga whale coronavirus SW1* (*Cegacovirus* subgenus), *Avian coronavirus, Avian coronavirus 9203*, and *Duck coronavirus 2714* (*Igacovirus* subgenus). The deltacoronaviruses include seven species: *Wigeon coronavirus HKU20 (Andecovirus* subgenus), *Bulbul coronavirus HKU11*, *Common Moorhen coronavirus HKU21*, *Coronavirus HKU15*, *Munia coronavirus HKU13, White-eye coronavirus HKU16 (Buldecovirus* subgenus), and *Night Heron coronavirus HKU19* (*Herdecovirus* subgenus) [1]. The variants of gamma- and deltacoronaviruses can infect not only birds, but pigs and whales as well [2,3]. Alpha- and betacoronaviruses are also known to infect bats, rodents, and humans, and SARS-CoV-2 (a *Betacoronavirus* species) is the virus that caused the recent and ongoing pandemic of COVID-19.

The genome of *Coronaviridae* species has a length between 27,317 and 31,357 nucleotides. Viral genomic RNA encodes ORF1a and ORF1b (translated to pp1a and pp1b), while the last third part of the RNA encodes a follow structural proteins spike (S), membrane (M), envelope (E), and nucleocapsid (N) [4].

Wild waterfowl are considered to be reservoirs of gammacoronaviruses. Most reported cases of this genus in wild birds have been asymptomatic, whereas *Coronaviridae* species are associated with mass die-offs among wild birds [5]. However, farmed poultry is endangered by *Gammacoronavirus* species (infectious bronchitis virus–IBV and the Turkey coronavirus [TCoV]), which cause economic losses in the poultry industry. Recombination has been shown for IBV and TCoV viruses [6].

First described in 1931, the IBV virus is of particular concern in the poultry industry, causing the suffocation and debilitation of chickens [7]. Despite the fact that vaccination is available and commonly practiced nowadays, IBV continues to cause outbreaks in poultry farms and is a major cause of significant economic loss [8].

The role of infected but asymptomatic wild migrating birds in the dissemination of viruses over long distances was first described at the beginning of the 1970s when avian influenza A viruses (AIV) and avian paramyxoviruses (subfamily *Avulavirinae*, family *Paramyxoviridae*) (APMV used hereafter for the purposes of this paper) in wild waterfowl were described in multiple publications, including active surveillance in Russia [9,10,11,12,13,14,15,16]. The probability of new zoonotic variants intruding into the human population raises concerns following the history of previous influenza pandemics and recent sporadic fatal cases of avian influenza in humans [17,18,19]. Previous pandemics were linked with AIVs, which circulated among wild birds, the segmented genome structure of AIV allowing the virus to gain new features with genetic shifts. Despite ACoVs not having the gene reassortment ability of AIVs, it has been shown that coronaviruses can have inter-subgenus recombinations [20], which can lead to changes in hosts.

Previous works have shown a significant presence of coronaviruses in wild bird populations. For example, they were shown to be present in 15.3% of infected birds in Australia, where both *Gamma*- and *Deltacoronavirus* variants were found [21]. For the Eurasian continent, a study from Poland showed a high prevalence of ACoVs over a long observational period (4.15%) [22]. With the exception of *Galliformes*, the most common order affected by gammacoronaviruses is thought to be *Anseriformes*, whereas deltacoronaviruses have a low prevalence in species of this order.

A vast area containing wild bird breeding sites in Russia remained untested for ACoV until our study and two independent investigations were carried out [23,24]. Research conducted in 2020 provided data on the presence of wild bird coronaviruses in 14 regions of Russia [23], while early study of Beringia area in 2010 revealed ACoVs in geese, ducks, and shorebirds [24]. Our study provides data on the presence of coronaviruses among wild birds in two observational sites in Russia in 2021, which represent two most significant key points among the breeding sites of birds using two principal Eurasian flyways. The first site is the Novosibirsk region in the southwest of Siberia, which is crossed by at least three migratory routes (west Asian–east African, central Asian, and Black Sea–Mediterranean flyways) which interconnect regions of Eurasia and Africa across long distances. The water bodies of the Novosibirsk region are breeding zones for many *Anseriformes* species [25]. Sakhalin Island is a migration site for *Anseriformes* and *Charadriiformes* of the East Asia–Australasian migratory flyway, which crosses Eurasia, North America, and Australia. Here, we found for the first time co-circulation and co-infection cases of avian coronavirus, avian paramyxovirus, and avian influenza A virus in wild duck species, which have their breeding areas in Siberia.

## 2. Materials and Methods

### 2.1. Ethics 

The present study was conducted in accordance with the approval and requirements of the Biomedical Ethics Committee of the Federal Research Center of Fundamental and Translational Medicine (FRC FTM), Novosibirsk (Protocols No. 2013-23 and 2021-10). The bird specimens were collected during the state hunting season with a license from the regional Ministries of Ecology and Natural Resources as part of the annual collection of biological material (the Programme for the Study of Infectious Diseases of Wild Animals, FRC FTM, Novosibirsk). The study utilised the Biosafety Level-3 (BSL-3) facilities of the FRC FTM.

### 2.2. Sample Collection

Cloacal swabs of wild waterfowl were collected during the hunting season in individual 2 mL tubes containing 1 mL of viral transport medium. The tubes containing sample biomaterial were stored in liquid nitrogen immediately and transported to the laboratory for analysis [26].

### 2.3. Avian Influenza Virus and Avian Paramyxovirus Isolation Using Chicken Embryos

Aliquots of each collected sample were used to isolate AIVs and APMVs. For this purpose, samples were mixed using a vortex shaker and transferred to new 1.5 mL tubes following centrifugation for 3 min at 3000 *g*. Supernatants were transferred to a new 1.5 mL tube containing penicillin and gentamicin. SPF chicken embryos (3 per sample) were inoculated with a 100 μL of sample in the allantoic cavity and incubated for 72 h in the BSL-3 laboratory of the FRC FTM [26]. Allantoic fluid was collected in individual tubes and tested for haemagglutinating activity. After 3 serial passages of virus cultivation, all HA-positive samples were aliquoted for AIV M gene PCR testing.

### 2.4. RNA Extraction, Reverse Transcription, and Real-Time PCR

#### 2.4.1. Avian Influenza Virus Detection

RNA was extracted from allantoic fluid samples using a kit for nucleic acid extraction (Medico-Biological Union LLC, MBU Group, Novosibirsk, Russia) following the manufacturer’s protocol. A measure of 5 μL of RNA was used to conduct RT-PCR with an AIV Real-Time RT-PCR kit (Medico-Biological Union LLC, MBU Group, Novosibirsk, Russia).

#### 2.4.2. Avian Paramyxovirus Detection

All samples of allantoic fluid with HA activity were also tested for the presence of viruses of the *Paramyxoviridae* family by PCR. For this purpose, RNA was isolated as described above and reverse transcription was performed using a Reverta-L kit (AmpliSens, Russia). To detect avian paramyxoviruses, family-wide oligonucleotides (PMX1 5′-GAR-GGI-YII-TGY-CAR-AAR-NTN-TGG-AC-3′ and PMX2 5′-TIA-YIG-CWA-TIR-IYT-GRT-TRT-CNC-C-3′) specific to domain III of the RNA-dependent RNA polymerase gene were used [27]. Oligonucleotides were diluted to a concentration of 50 pmol/μL. A reaction mixture was prepared using 25 μL of Quick-Load Taq 2x Master Mix (New England Biolabs, USA), 1 μL of forward and reverse oligonucleotides, and 5 μL of cDNA. Water was then added to achieve a final volume of 50 µL. The reaction mixture was incubated at 94 °C for 1 min, then for 40 cycles at 94 °C for 15 s, at 41 °C for 30 s, and at 68 °C for 30 s, and then a final extension at 68 °C for 7 min.

Reaction products were visualised by 1.5% agarose gel electrophoresis in Gel Doc XR+ (Bio-Rad Laboratories, Hercules, CA, USA). A 100 bp DNA Ladder O’GeneRuler Plus (Thermo Fisher Scientific, USA) was used to estimate amplicon size. Samples in which amplicons were found were prepared for whole-genome sequencing.

#### 2.4.3. Avian Coronavirus Detection and Fragment Sequencing

RNA was extracted from aliquots of cloacal swab samples using a kit for nucleic acid extraction (Medico-Biological Union LLC, MBU Group, Novosibirsk, Russia) following the manufacturer’s instructions. IBV vaccine strain H120 was used as a positive control. The protocol for ACoV detection using modified family-wide oligonucleotides [21] for 602 nucleotide fragments of RdRp of ACoV was implemented. Reverse transcription using 100 pmol of reverse oligonucleotide, 4 μL of RT buffer, 1 μL of reverse transcriptase, and 10 μL of RNA was implemented using a RNAScribe kit (Biolabmix, Novosibirsk Region, Russia) in the following conditions: 50 °C for 40 min, 85 °C for 5 min. PCR with a SYBR Blue HS-qPCR kit (Biolabmix, Novosibirsk Region, Russia) was carried out mixing 1 μL of H_2_O, 5 μL of BiomasterMix, 1 pM forward, and 1 pM reverse oligonucleotides. PCR in the following conditions was implemented: at 95 °C for 30 s, annealing for 30 s and at 72 °C for 45 s following final elongation at 72 °C for 3 min. Annealing temperature decreased every 3 cycles by 2 °C from 60 °C to 48 °C. The main phase at the 48 °C annealing temperature had 30 cycles. Melting curves were constructed according to the following conditions: at 95 °C for 15 s, at 60 °C for 1 min, and at 60 °C to 95 °C, with 0.05 °C/s increments.

To visualise and detect PCR products, we used electrophoresis in 1.5% agarose gel, and detected amplicons were sliced from the gel and extracted using a GeneJet Gel Extraction kit (Thermo Fisher Scientific, Waltham, MA, USA), following the manufacturer’s instructions. Extracted amplified DNA was used for the sequencing reaction with a BigDye V3.1 kit (Thermo Fisher Scientific, Waltham, MA, USA). Fragments were sequenced using an ABI 3130XL Genetic Analyser (Applied Biosystems, Waltham, MA, USA) in accordance with the manufacturer’s instructions at the Genomics Core Facility of the Siberian Branch of the Russian Academy of Sciences (ICBFM SB RAS, Novosibirsk, Russia).

### 2.5. Phylogenetic Analysis

To determine the genus of positive samples, phylogenetic analysis was conducted. The most relevant and reference sequences from the NCBI GenBank database were added for analysis. Sequences were aligned using a MUSCLE algorithm in MEGA X. A maximum likelihood phylogenetic tree was generated using the GTR + G + I substitution model with a bootstrap test, 1000 iterations.

## 3. Results

### 3.1. Virus Detection

We collected 606 samples from 12 species of hunt-harvested wild ducks from two sites of the Asian part of Russia: the Novosibirsk region in the Western Siberian Lowland wetlands (n = 389) and Sakhalin Island (n = 217) (Table 1). We constructed a map showing both exact sampling sites of this study, and the Gammacoronavirus detections from the other available studies (Figure 1).

We evaluated samples for the presence of three viruses and their mixed infection using PCR. We found that the most prevalent virus in single-infected samples was avian influenza A virus, n = 37 (6.1%) out of 606 samples, followed by avian paramyxovirus, n = 33 (5.4%), and avian coronavirus, n = 25 (4.1%). We also found 14 samples simultaneously infected with two viruses in the following combinations: AIV + ACoV (n = 8, 1.3%), APMV + ACoV (n = 5, 0.8%), and AIV+ APMV (n = 1, 0.2%) (Figure 2). One sample from a Northern Pintail (*Anas acuta*) collected on Sakhalin Island was found to be positive for three infections: AIV+ ACoV+ APMV, which is a 0.2% proportion of the 606 study samples analysed. Finally, the total number of individuals infected with any virus was found to be 110 (16.6%). The isolation rate was shown to be 15.9% and 22.1% for the Novosibirsk region in the Western Siberian Lowland wetlands (n = 62) and Sakhalin Island (n = 48), respectively.

The largest number of all positive samples was detected among *Anas crecca* (n = 51) and *Anas acuta* (n = 12). *Coronavirus* RNA in the form of single infections was found in samples from the following species: *Anas crecca* (n = 8), *Anas stepera* (n = 5), *Anas formosa* (n = 3), *Anas acuta* (n = 2), and some other species (n = 7).

A single infection was found in all species studied except for the Red-crested Pochard (*Netta rufina*) and Goldeneye (*Bucephala clangula*). Co-infection positives were only found in the three most represented species, which had large sample sizes: Common Teal (*Anas crecca*), Northern Pintail (*Anas acuta*), and Gadwall (*Anas strepera*), the latter species having only one combination, avian paramyxovirus and avian coronavirus. Combinations with APMV were more prevalent in the samples from Sakhalin Island (n = 5) than in those from the Novosibirsk region (n = 1).

Thus, we found 25 avian coronaviruses in the form of a single infection (4.1%) and another 14 in the form of co-infections with other avian viral infections (2.3%).

### 3.2. Phylogenetic Analysis of Coronaviruses

We obtained nucleotide sequences of the RdRp fragments for 16 coronaviruses isolated from the Sakhalin Island samples and for 6 coronaviruses isolated from the Western Siberian Lowland samples (GenBank accession numbers OQ731809-OQ731830). Other isolates were sequenced and assigned *Gammacoronaviruses*, but were not subjected to phylogenetic analysis because the quality and length of the sequences were insufficient. Phylogenetic analysis revealed that all samples obtained were related to *Gammacoronavirus* (Figure 3). Figure 3A shows the position of the studied strains relative to the reference strains of four genera of coronaviruses: Alphacoronaviruses (seasonal human coronaviruses), Betacoronaviruses (SARS-CoV, MERS-CoV, SARS-CoV-2, Human seasonal OC43 and HKU1), Gammacoronaviruses, and Deltacoronaviruses. The tree topologies (Figure 3B) did not show significant clustering between viruses of different regions or hosts. When constructing a phylogenetic tree of amino acid sequences (103 a.a., only *Gammacoronavirus* included, tree is not shown), we showed that all our sequences belong to a *Duck coronavirus* species and no longer have such a complex topology as at the nucleotide phylogenetic tree. The visualization of the pairwise distances of the nucleotide sequences confirmed a significant difference in the studied gammacoronaviruses from known zoonotic coronaviruses (Appendix A).

## 4. Discussion

Among the 606 cloacal swab samples collected from wild waterfowl at two distant ornithological surveillance hotspots in 2021, we detected 39 coronaviruses. We found that 25 avian coronaviruses (i.e., 4.1%) out of the total number of samples were in the form of a single infection, and another 14 (i.e., 2.3%) were in the form of co-infection with other tested avian viral infections.

The most common coronavirus was found in co-infection with influenza, followed by paramyxoviruses. Additionally, for the first time, we found a wild bird, a Northern Pintail, on Sakhalin Island that simultaneously had three infections: AIV + ACoV + APMV. The Northern Pintail (*Anas acuta*) is a wide-ranging migratory duck with a Holarctic breeding and wintering distribution and is a major vector for intercontinental virus exchange and movement of avian influenza A viruses [28].

It can be assumed that there is an exchange of coronaviruses between Eurasia and America which has not yet been documented in the published literature. However, in North America, coronaviruses in wild migrating birds have been found in very limited numbers, which suggests that waterfowl and shorebirds are not significant natural reservoirs for ACoVs in North America, although sample size, collection method, collection location, and bird age may have impacted the available prevalence data [29]. However, for some shorebird species, there was shown to be a high prevalence of gammacoronaviruses [30]. Our phylogenetic analysis did not show a close relationship with American coronaviruses. On the other hand, we found that the isolate ACoV/Gadwall/Sakhalin/213 clustered with some ACoVs detected in North America and Australia (2016) (Figure 3), suggesting the possibility of virus exchange in Beringia. However, bootstrap indices are not enough for a clear conclusion. A low prevalence of coronavirus was also found in some studies in urbanised areas in South America [31]. However, none of the above-mentioned studies investigated the co-circulation of the main viruses for which monitoring is undertaken and which we have identified in Siberia through the present research.

Along with a single coronavirus infection (4.1%), we found that 1.3% of samples were co-infected with ACoV and AIV. For the first time, we showed co-infection with avian coronavirus and paramyxovirus (0.8%). Previous research in China revealed co-infections of ACoV and AIV and that the presence of co-infected birds was relatively high (3.3%) [32]. The highest proportion of individual birds with multiple co-infections in our study was shown for Common Teals (*Anas crecca*) (n = 51, 28.3%), which had the highest diversity of combinations. However, it should be noted that this species constituted the largest number in terms of sample size. Earlier, our research revealed a high percentage of influenza A viruses and avian paramyxoviruses, but ACoV was not included in the studies [13,14,15,16].

In the present study, only three viruses (one paramyxovirus and two coronaviruses) from 110 samples were detected in the diving ducks: Common Pochard (*Aythya ferina*); Red-crested Pochard (*Netta rufina*), Tufted Duck (*Aythya fuligula*), and Goldeneye (*Bucephala clangula*), which confirms the role of dabbling ducks as principal reservoirs for the viruses studied. In addition, the greatest number of co-infections and their combinations was found in only two species of dabbling ducks, the Common Teal and Northern Pintail.

All partially sequenced coronaviruses were assigned to *Gammacoronavirus*: we did not detect deltacoronaviruses. On the one hand, these data support the assumption of the low prevalence of deltacoronaviruses among wild birds, because we used primers that amplified both gamma- and deltacoronaviruses with equal efficiency. On the other hand, this result for coronaviruses in general may be biased by the detection method implemented in our study, whereby we tested for coronavirus from the original swabs collected, but for AIVs and paramyxoviruses from cultivated chicken embryos. The different methods of investigating samples may have led to differing virus prevalence estimates, which places a limitation on our effective comparative analysis. Our study indeed has this limitation, as avian coronaviruses do not grow well in cells and chicken embryos, and thus we used PCR for their detection, while for avian influenza and paramyxoviruses, we used chicken embryos, which are more responsive to these viruses. We aimed to show whether we could detect different viruses in our samples, rather than compare the impact of the methods employed on the results. The 2022 study by Marchenko et al. of samples taken from wild waterfowl of the Novosibirsk region revealed only one deltacoronavirus in a Gadwall collected in 2020, while the phylogeny of gammacoronaviruses and their hosts were revealed as miscellaneous [23]. Nevertheless, similar studies in wintering sites in China revealed a large prevalence of deltacoronaviruses [32]. At the same time, gammacoronaviruses were also found there, mainly in wild ducks, whereas deltacoronaviruses were found mainly among the *Ciconiiformes* and *Columbiformes*, which are the dominant resident bird orders in wintering places in eastern China (Shanghai), but they were not represented in our studies. Thus, our study, together with data from other studies, supports the assumption that wild ducks may not play a key role as a reservoir of Deltacoronaviruses.

If we look at birds migrating to the north of such wintering sites in eastern China and Korea, along the East Asia–Australia migratory flyway, only gammacoronaviruses were detected in the Northern Pintail and Indian Spot-billed Duck [33]. However, did not find phylogenetically similar viruses on Sakhalin Island.

It should be noted that the samples in this study were collected during the autumn migrations of these species to wintering areas. The key sampling points were in various territories located on differing migration routes. The Novosibirsk region populations mainly migrate in a southwestward direction to wintering sites in central Asia, the Black Sea, and the Mediterranean basin [25]. As this kind of research has not been conducted in the European territories within the migration routes of the species sampled in our study, it is, as yet, impossible to compare data on the co-circulation of the viruses detected. However, studies in Europe have revealed the diversity of coronaviruses in wild birds, at least in their nesting and permanent habitats. For instance, a study in Poland detected gammacoronaviruses (Figure 1) more often than deltacoronaviruses, with detection rates of 3.5% and 0.7%, respectively, while the total prevalence of coronaviruses revealed in wild bird populations was 4.15%, and the main viral reservoirs were amongst birds of the orders *Anseriformes* and *Charadriiformes* [22]. A similar pattern was found in Portugal, with the detection of only one deltacoronavirus infection (1.4%) against the background of a high prevalence of detected gammacoronaviruses (31.4%) [34]. At the same time, there has been no study on viral co-infections in birds in Europe, with the exception of a detailed significant work on three infections in wild Mallards [35], although the probability of such co-infections in European wild birds would seem to be high. Such work is necessary for the development of monitoring systems for multiple bird infections, for example, similar to those that were created for the highly pathogenic avian influenza [36].

From the partial sequences we obtained of our coronaviruses, we constructed a phylogenetic tree by comparing the partial sequences of the RdRp with those in the GenBank database (Figure 3). The results showed that gammacoronaviruses are mainly clustered with duck and shorebird ACoVs mainly found in different countries of Europe and Asia, including viruses currently found in Siberia [16]. One isolate had relation to the cluster of American and Australian wild duck ACoVs.

We did not find close phylogenetic relationships of our coronaviruses with the strains in southern wintering areas mentioned above. Unfortunately, there are few studies on the comparison of coronaviruses and their co-infections in wintering sites.

In summary, the present study demonstrated that ACoV, AIV, and APMV co-infection is highly prevalent in wild birds, including some cases of triple co-infection.

The long-term continual surveillance of such co-infections in wild birds in their breeding and wintering areas is required to better understand the ecology and epidemiology of these viruses.

## 5. Conclusions

The present study demonstrated that ACoV infection is highly prevalent in wild migrating ducks of dabbling species, not only as single infections or co-infections with avian influenza virus, but also with paramyxoviruses. We first detected the triple co-infection of these viruses in a long-distance migrating species, the Northern Pintail.

These data are essential for the fundamental understanding of the diversity and dynamics of ACoV in wild bird populations in association with other zoonotic avian viruses, such as avian influenza viruses and avian paramyxoviruses.

## Figures and Tables

**Figure 1 viruses-15-01121-f001:**
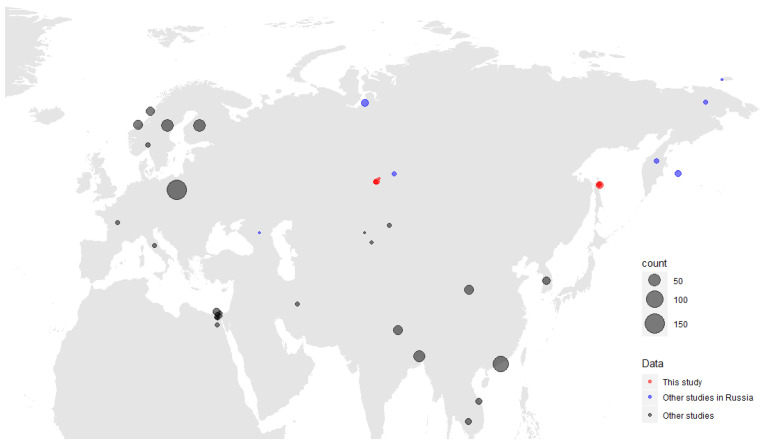
Map of Gammacoronaviruses detected in this study (red). Detections from other available studies are mapped according to regions, and also assigned to a different colour category (blue for Russian studies, black for other Eurasian and African studies); only viruses from wild birds are presented. A size scale shows the number of viruses detected.

**Figure 2 viruses-15-01121-f002:**
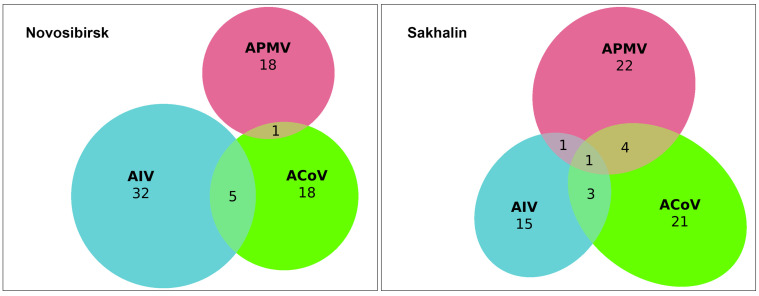
Euler diagram showing number of avian influenza A virus (AIV, blue color), avian coronavirus (ACoV, green color), and avian paramyxovirus (APMV, red color) infections and co-infections detected in wild birds in the Novosibirsk (left diagram) and Sakhalin regions (right diagram).

**Figure 3 viruses-15-01121-f003:**
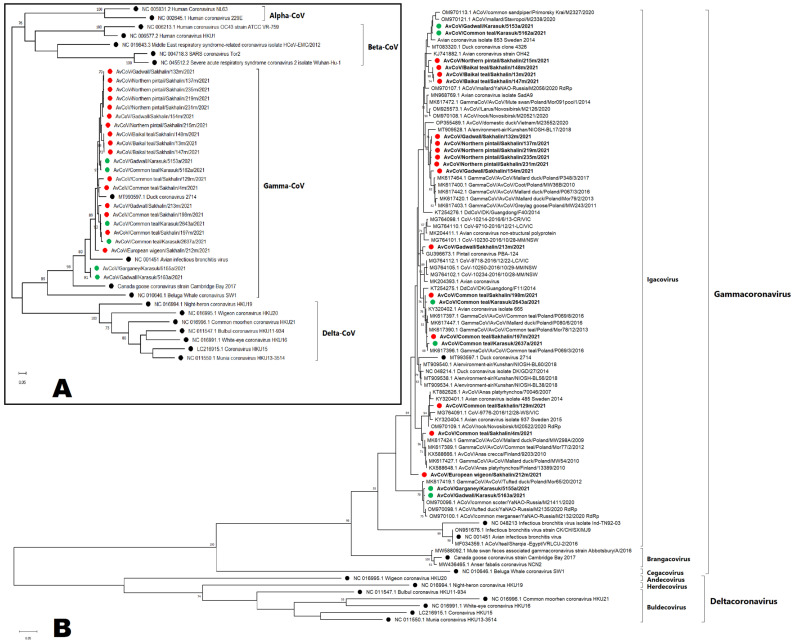
Maximum-likelihood phylogenetic tree of nucleotide sequences of RdRp fragments of four genera of coronaviruses: Alphacoronaviruses, Betacoronaviruses, Gammacoronaviruses, and Deltacoronaviruses (**A**). Detailed maximum-likelihood phylogenetic tree of nucleotide sequences of RdRp fragments of Gammacoronavirus and Deltacoronavirus. (**B**). Coronaviruses isolated from Sakhalin Island samples are marked red (●); coronaviruses isolated from the Western Siberian Lowland are marked green (●); reference strains are marked black (●).

**Table 1 viruses-15-01121-t001:** Sample size and results of virus detection of wild ducks in the Asian part of Russia.

Species	Number of Samples	Single Infection Positives			Co-Infection Positives				Total Number of Infected Individuals and %Age
		Avian Influenza (AIV)	Avian Coronavirus (ACoV)	Avian Paramyxovirus (APMV)	AIV + ACoV	AIV + APMV	APMV + ACoV	AIV + ACoV + APMV	
Common Teal (*Anas crecca*)	180	20	8	13	7	1	2	-	51 (28.3%)
Garganey (*Anas querquedula*)	37	3	1	1	-	-	-	-	5 (13.5%)
Mallard (*Anas platyrhynchos*)	47	3	1	6	-	-	-	-	10 (2.2%)
Northern Shoveler (*Anas clypeata*)	40	2	2	-	-	-	-	-	4 (10%)
Baikal Teal (*Anas formosa*)	38	1	3	7	-	-	-	-	11 (28.9%)
Northern Pintail (*Anas acuta*)	89	5	2	1	1	-	2	1	12 (13.5%)
Gadwall (*Anas strepera*)	81	1	5	4	-	-	1	-	11 (13.6%)
Wigeon (*Anas penelope*)	27	2	1	-	-	-	-	-	3 (11.1%)
Common Pochard (*Aythya ferina*)	40	-	1	1	-	-	-	-	2 (5%)
Red-crested Pochard (*Netta rufina*)	11	-	-	-	-	-	-	-	0
Tufted Duck (*Aythya fuligula*)	7	-	1	-	-	-	-	-	1 (14.2%)
Goldeneye (*Bucephala clangula*)	9	-	-	-	-	-	-	-	0
Novosibirsk Region sampling site	389	27	12	17	5	0	1	0	62 (15.9%)
Sakhalin Island sampling site	217	10	13	16	3	1	4	1	48 (22.1%)
Total	606	37 (6.1%)	25 (4.1%)	33 (5.4%)	8 (1.3%)	1 (0.2%)	5 (0.8%)	1 (0.2%)	110 (16.6%)

## Data Availability

All sequences from the study are available in GenBank (accession numbers: OQ731809-OQ731830).

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
