# Peer review of "Does Avian Coronavirus Co-Circulate with Avian Paramyxovirus and Avian Influenza Virus in Wild Ducks in Siberia?"

_viruses, 2023, doi:10.3390/v15051121_

Round 1

Reviewer 1 Report

In the reviewed manuscript, the prevalence of avian coronavirus (ACoV), avian paramyxovirus (APMV) and avian influenza A virus (AIV) in wild duck species of Siberia was investigated. The 606 cloacal swab samples were collected during the autumn migrations of birds to wintering areas, and a high presence of ACoV was established. ACoV was detected both as a single infection and as a co-infected with APMV and AIV.    Phylogenetic analysis revealed that all detected coronaviruses belonged to the genus Gammacoronavirus; Deltacoronaviruses were not detected. A comparative analysis of the prevalence of viruses among different species of waterfowl was carried out. It was concluded that the dabbling ducks as principal reservoirs for the tested viruses. The prevalence of viruses in the diving ducks was significantly lower. This work expands our knowledge of the circulation of potentially dangerous viruses in wild birds.

             Comments in attached file

Author Response

Dear Reviewers,

Thank you for your useful comments and constructive suggestions which helped us to improve the manuscript. Additional information was provided for results and discussion according to the comments. We edited text according to comments and suggestions. Please see below detailed response on each point of the review and corrections of the manuscript.

Reviewer 1.

In the reviewed manuscript, the prevalence of avian coronavirus (ACoV), avian paramyxovirus (APMV) and avian influenza A virus (AIV) in wild duck species of Siberia was investigated. The 606 cloacal swab samples were collected during the autumn migrations of birds to wintering areas, and a high presence of ACoV was established. ACoV was detected both as a single infection and as a co-infected with APMV and AIV.    Phylogenetic analysis revealed that all detected coronaviruses belonged to the genus GammacoronavirusDeltacoronaviruses were not detected. A comparative analysis of the prevalence of viruses among different species of waterfowl was carried out. It was concluded that the dabbling ducks as principal reservoirs for the tested viruses. The prevalence of viruses in the diving ducks was significantly lower. This work expands our knowledge of the circulation of potentially dangerous viruses in wild birds.

Some notes to the text are listed below: (My comments are in blue)

  • line 129: (MBS, Russia)                                                                              The abbreviation MBS (MBS, Russia) must be deciphered as it is not very well known

Response: We detailed information: (Medico-Biological Union LLC, MBU Group, Novosibirsk, Russia).

  • line 197: Figure Euler diagram of avian influenza A virus (AIV)…                Figure 1 is erroneously labeled as Figure 2

Response: We corrected

  • Line 204: samples from the following species: Anas crecca (n=8), Anas stepera (n=5), Anas formosa (n=3), Anas acuta (n=2), and from all other species (n=7).

        It's probably better to write some species, not all species.

Response: Corrected according to the comment

  • Lines 219-225:  After the discussion of the phylogenetic tree of nucleotide sequences, a description and analysis of a phylogenetic tree of amino acid sequences follows, but the tree itself is not given, instead a link is given to Fig 2, i.e. tree of nucleotide sequences.

Response: corrected

  • Lines 273-279:   The possibility that absent of detection of deltacoronaviruses is associated with different methods of virus isolation is discussed. However, this applies to the comparative detection of ACoVs, AIVs and APMVs. The detection of gammacoronavirus and deltacoronaviruses was the same. (At this point it would be good to recall that the chosen primers amplified these viruses with equal efficiency)

Response: Thank you! We agree – actually this paragraph was written inconsistently. We tried to correct this, we mentioned “we used primers that amplified both gamma- and deltacoronaviruses with equal efficiency”

  • Discussion: Missing of detection of deltacoronaviruses is discussed many times, but along the way it becomes clear that all detected Gammacoronaviruses belong to the genus of duck viruses and are isolated from duck birds, and the hosts of deltacoronaviruses are completely different birds (of other orders). This should probably be clearly stated in the article.

Response:

Actually, the host of deltacoronaviruses can also be ducks. We cannot say host species completely different. For example, one reference strain of delta-CoV we showed in the tree is “Wigeon coronavirus” isolated from common dabbling duck Mareca penelope.

Thus, we had a chance to detect delta-CoV, but we haven’t. Additionally, Marchenko et. al detected deltacoronavirus from wild duck – Anas strepera in Asian part of Russia.

We add the conclusion sentence that our study, together with data from other studies, confirm that wild ducks may not play a key role as a reservoir of Deltacoronaviruses.

Additionally:

  1. We divided the figure into two diagrams – for two different regions
  2. We tried to add some information. Unfortunately, we could not measure antibody levels as we did not have a chance to take blood from hunt-harvested birds. Also we did not provide experiments for quantitative analysis of virus shedding. We believe this is the aim for different study.
  3. We added a map with our viruses sampled and all detections from other available studies are mapped according to regions, and also assigned to a different color category (blue- for Russian studies, black for other Eurasian studies). A size scale shows the number of viruses detected.
  4. We changed the tree and added the heat map – please see response to the specific comment below
  5. We added some discussion of phylogenetic relations, including clustering with American viruses, however without strong bootstrap support

We put 4 additional references

Reviewer 2 Report

This article by Sharshov et al. shows for the first time the co-infection of avian paramyxovirus+ avian influenza virus+ avian coronavirus in Siberian wild ducks. Detection of these viruses were done by collecting cloacal swabs of wild ducks from the Sakhalin and Novosibirsk regions of Russia. The article is well-organized and the findings are extensively discussed in the discussion section. Here are some suggestions to improve this publication:

Major comments:

Figure 1: It will be more informative to divide this diagram into the two regions from where sample was collected.

According to me, some more experiments or bioinformatic analysis could have been added, especially because the phylogenetic analysis provided here was inconclusive. For eg. there could be a figure depicting viral titers in each avian species; the serum antibody levels of these birds could be measured; mapping the sampling location used in this study and pointing out locations which has been used in similar previous studies from Asian region of Russia could be a supplementary figure; virus diversity in each location related analysis could be done to dive deeper into Fig 2; connection/similarity of these detected viruses (mentioned in phylogenetic tree) with viruses infecting humans could be analyzed; there could be prediction analysis or at least speculation of those viruses in phylogenetic tree to become zoonotic. And these are just some suggested examples- exactly all of this need not be performed.

Minor comments:

Line 44: Please expand "ICTV".

Line 57: "S, N, E, M" - these should be either expanded or removed from the text.

Line 72: "multiple publications [9]" - it will be better to cite some of those publications instead of a review.

Line 88: Kindly add references of the "two independent investigations".

Fig 2: needs to be expanded (maybe a full page). Was there any other relationship/interaction/observations noticed in this phylogenetic tree which can be added and explained in the results section? References 15 (Domanska-Blicharz et al, 2021) and 22 (Canuti et al, 2019) can be used as example as those studies have meticulously analyzed their phylogenetic results.

The quality of language is satisfactory. There is inconsistent formatting of virus names in multiple places throughout the manuscript. Also, semicolon are used in multiple places throughout the manuscript in place of comma.

Author Response

Dear Reviewers,

Thank you for your useful comments and constructive suggestions which helped us to improve the manuscript. Additional information was provided for results and discussion according to the comments. We edited text according to comments and suggestions. Please see below detailed response on each point of the review and corrections of the manuscript.

Reviewer 2.

Comments and Suggestions for Authors

This article by Sharshov et al. shows for the first time the co-infection of avian paramyxovirus+ avian influenza virus+ avian coronavirus in Siberian wild ducks. Detection of these viruses were done by collecting cloacal swabs of wild ducks from the Sakhalin and Novosibirsk regions of Russia. The article is well-organized and the findings are extensively discussed in the discussion section. Here are some suggestions to improve this publication:

Major comments:

Figure 1: It will be more informative to divide this diagram into the two regions from where sample was collected.

According to me, some more experiments or bioinformatic analysis could have been added, especially because the phylogenetic analysis provided here was inconclusive. For eg. there could be a figure depicting viral titers in each avian species; the serum antibody levels of these birds could be measured; mapping the sampling location used in this study and pointing out locations which has been used in similar previous studies from Asian region of Russia could be a supplementary figure; virus diversity in each location related analysis could be done to dive deeper into Fig 2; connection/similarity of these detected viruses (mentioned in phylogenetic tree) with viruses infecting humans could be analyzed; there could be prediction analysis or at least speculation of those viruses in phylogenetic tree to become zoonotic. And these are just some suggested examples- exactly all of this need not be performed.

Response:

  1. We divided the figure into two diagrams – for two different regions
  2. We tried to add some information. Unfortunately, we could not measure antibody levels as we did not have a chance to take blood from hunt-harvested birds. Also we did not provide experiments for quantitative analysis of virus shedding. We believe this is the aim for different study.
  3. We added a map with our viruses sampled and all detections from other available studies are mapped according to regions, and also assigned to a different color category (blue- for Russian studies, black for other Eurasian studies). A size scale shows the number of viruses detected.
  4. We changed the tree and added the heat map – please see response to the specific comment below
  5. We added some discussion of phylogenetic relations, including clustering with American viruses, however without strong bootstrap support
  6. We put 4 additional references

Minor comments:

Line 44: Please expand "ICTV".

Response: corrected

  1. Line 57: "S, N, E, M" - these should be either expanded or removed from the text.

Response: We expanded.

  1. Line 72: "multiple publications [9]" - it will be better to cite some of those publications instead of a review.

Response: we added some historical and current studies.

  1. Line 88: Kindly add references of the "two independent investigations".

Response: corrected. "two independent investigations" - we meant independent investigation carried out by Marchenko et al. [16]. And we added second one : Muradrasoli S, Bálint A, Wahlgren J, Waldenström J, Belák S, Blomberg J, Olsen B. Prevalence and phylogeny of coronaviruses in wild birds from the Bering Strait area (Beringia). PLoS One. 2010 Oct 29;5(10):e13640. doi: 10.1371/journal.pone.0013640.

  1. Fig 2: needs to be expanded (maybe a full page). Was there any other relationship/interaction/observations noticed in this phylogenetic tree which can be added and explained in the results section? References 15 (Domanska-Blicharz et al, 2021) and 22 (Canuti et al, 2019) can be used as example as those studies have meticulously analyzed their phylogenetic results.

Response: We changed the figure dramatically, added common tree four genera of coronaviruses: Alphacoronaviruses, Betacoronaviruses, Gammacoronavirus and Deltacoronavirus (A). and modified detailed Maximum likelihood phylogenetic tree of nucleotide sequences of RdRp fragments of Gammacoronavirus and Deltacoronavirus (B) , where we added subgenera. Also we added Heatmap with visualization of pairwise distances of nucleotide sequences of studied gammacoronaviruses as a additional Supplementary to show distance from zoonotic coronaviruses.

  1. Comments on the Quality of English Language

The quality of language is satisfactory. There is inconsistent formatting of virus names in multiple places throughout the manuscript. Also, semicolon are used in multiple places throughout the manuscript in place of comma.

Response: Thank you! We tried to change “semicolon” to com